# The Prospects of Secondary Moderate Mitral Regurgitation after Aortic Valve Replacement —Meta-Analysis

**DOI:** 10.3390/ijerph17197335

**Published:** 2020-10-08

**Authors:** Ilija Bilbija, Milos Matkovic, Marko Cubrilo, Nemanja Aleksic, Jelena Milin Lazovic, Jelena Cumic, Vladimir Tutus, Marko Jovanovic, Svetozar Putnik

**Affiliations:** 1Department for Cardiac Surgery, Clinical Center of Serbia, 11000 Belgrade, Serbia; dr.matko@hotmail.com (M.M.); cubrilomarko@gmail.com (M.C.); ner.vuk@hotmail.com (N.A.); svetozar073@yahoo.com (S.P.); 2Department of Surgery with Anesthesiology, Faculty of Medicine, University of Belgrade, 11000 Belgrade, Serbia; 3Department for Biostatistics, Faculty of Medicine, University of Belgrade, 11000 Belgrade, Serbia; milinjelena@gmail.com; 4Department for Anesthesiology and Intensive Care, Clinical Center of Serbia, 11000 Belgrade, Serbia; jelena.cumic@gmail.com (J.C.); vltutus@yahoo.com (V.T.); 5Institute for Cardiovascular Diseases Dedinje, 11000 Belgrade, Serbia; jovanovicmvma@yahoo.com

**Keywords:** mitral regurgitation, aortic valve replacement, aortic stenosis, combined aortic mitral surgery

## Abstract

Aortic valve replacement for aortic stenosis represents one of the most frequent surgical procedures on heart valves. These patients often have concomitant mitral regurgitation. To reveal whether the moderate mitral regurgitation will improve after aortic valve replacement alone, we performed a systematic review and meta-analysis. We identified 27 studies with 4452 patients that underwent aortic valve replacement for aortic stenosis and had co-existent mitral regurgitation. Primary end point was the impact of aortic valve replacement on the concomitant mitral regurgitation. Secondary end points were the analysis of the left ventricle reverse remodeling and long-term survival. Our results showed that there was significant improvement in mitral regurgitation postoperatively (RR, 1.65; 95% CI 1.36–2.00; *p* < 0.00001) with the average decrease of 0.46 (WMD; 95% CI 0.35–0.57; *p* < 0.00001). The effect is more pronounced in the elderly population. Perioperative mortality was higher (*p* < 0.0001) and long-term survival significantly worse (*p* < 0.00001) in patients that had moderate/severe mitral regurgitation preoperatively. We conclude that after aortic valve replacement alone there are fair chances but for only slight improvement in concomitant mitral regurgitation. The secondary moderate mitral regurgitation should be addressed at the time of aortic valve replacement. A more conservative approach should be followed for elderly and high-risk patients.

## 1. Introduction

Aortic valve replacement (AVR) aimed to relieve the aortic stenosis (AS) represents one of the most commonly undertaken surgical procedures on heart valves [1,2]. In clinical practice it is very common to see patients with severe calcific aortic stenosis who have concomitant functional mitral regurgitation (MR) of a certain degree [3].

Mitral insufficiency accompanied with aortic stenosis adversely affects both morbidity and mortality of these patients [3,4,5]. If the mitral insufficiency is severe, according to current guidelines outlined by the American Heart Association (AHA) and the European Society of Cardiology (ESC) and the European Association for Cardio-Thoracic Surgery (EACTS), double valve surgery is recommended [6]. On the other hand, if mitral regurgitation is mild, it should be left as it is.

Double valve surgery (aortic and mitral) carries a significantly higher operative risk comparing to aortic valve replacement alone [3,7]. Perioperative mortality of these procedures reaches 11–12%. In contrast, the data from the Society of Thoracic Surgery (STS) database reports overall unadjusted mortality of 3.2% for AVR alone. In the long term, the rate of thromboembolic and hemorrhagic complications is also elevated. This applies to the incidence of endocarditis as well. For cases of aortic stenosis accompanied with MR of a moderate degree there are no clear treatment guidelines [6].

There are a number of observational studies already published on this issue, but none of them was able to explain the fate of moderate MR after AVR for AS and its impact on survival [7]. The quantitative analysis of the pooled data could provide the answers and therefore possibility to create a treatment strategy for this subgroup of patients.

Primary endpoint of this meta-analysis is the impact of AVR on the secondary mitral regurgitation. Secondary endpoints include the analysis of the scope of reverse remodeling, periprocedural mortality and long-term survival of these patients.

## 2. Methods

### 2.1. Literature Search

A literature search was performed according to PRISMA strategy [8] through PubMed, Scopus and WOS (Web of science) databases with keywords “aortic valve replacement” AND “mitral regurgitation” OR (aort* valv* replac* and mitr* regurgit*). The search was conducted by two authors independently. In addition, the search was broadened through databases of peer-reviewed journals in the field of cardiac surgery. After removing duplicates, all records were screened to find relevant studies. There was no restriction in terms of language or publication date. The last date of the search was 31 May 2019. The references of relevant articles were searched manually to find any studies that were inadvertently omitted.

### 2.2. Inclusion and Exclusion Criteria

After obtaining full text versions of all relevant articles, they were assessed for eligibility. The inclusion criteria were studies reporting patients that underwent isolated, first time surgical aortic valve replacement for aortic stenosis with concomitant secondary MR. The reasons for exclusion of studies were operative procedures on mitral valve, aortic insufficiency as the indication for AVR, paucity or inconsistency of data prohibiting valid extraction. If the study included different groups of patients or treatments, we included the arms that met inclusion criteria if it was appropriate. Any disagreement between the two authors searching for studies was solved through consensus.

### 2.3. Data Extraction

The authors extracted following data from eligible studies: first author, year of publication, type of study, number and characteristics of participants, clinical status, comorbidities, preoperative and postoperative echocardiography parameters, type of AVR and all outcome measures reported. Specifically, we focused to extract outcome data where possible for the degree and change of MR severity pre and postoperatively, change in the ejection fraction (EF) of the LV, change in the left ventricle end diastolic diameter (LVEDD) and mitral annulus diameter, 30-day mortality and long-term survival.

### 2.4. Risk of Bias

The possible risk of bias was estimated following Cochrane collaboration recommendations. Each study was individually assessed for the selection bias (random sequence generation and allocation concealment), performance bias—blinding of participants and personnel, detection bias—blinding of outcome assessment, attrition bias—incomplete outcome data and selective reporting—reporting bias. For each of these biases the estimated risk was recorded as high risk, low risk or unclear risk. The publication bias was evaluated through the construction of a funnel plot graph.

### 2.5. Data Analysis

All subjects in the review underwent the same treatment—AVR. There was no control group for comparison. Where possible, we compared the parameters before and after the treatment and pooled data for more precise estimate of mortality and long-term survival. The meta-analysis was performed following recommendations from the Cochrane collaboration guidelines [9]. All details about studies are stored on the OSF (https://osf.io/3ybeq/?view_only=9e9bfcec4dd443efb919a258fb490edf).

The data for MR severity pre and postoperatively was pooled from individual subjects’ data across studies and analyzed as risk ratio (RR) using a random effects or fixed effects model as appropriate with MH method. The publications were divided into two subgroups according to the average patient age. The cut-off value was set at 70 years. Subgroup analysis for younger and older patients was conducted for severe MR and moderate and severe MR combined. Inter-study heterogeneity was explored using chi² statistics and calculating I² value to quantify the degree of heterogeneity. I^2^ represents the inconsistency between study results and quantifies the proportion of observed dispersion that is real, i.e., that is due to between-study differences and not to random errors. When I² was more than 50% it was considered that significant heterogeneity was present.

Preoperative and postoperative echo parameters were analyzed as weighted mean difference (WMD) through random effects model with 95% confidence intervals (CI). WMD was estimated by pooling individual trial results using random-effects models via the Der Simonian-Laird method. A separate forest plot was constructed for each analysis showing the WMD (box), 95% CI (lines) and weight (size of box) for each trial. The overall effect size is represented by a diamond. A positive WMD favored the reduction in magnitude of the observed variable following AVR. The point of estimate of WMD was considered statistically significant if *p* < 0.05 and 95% confidence interval did not include zero. The inverse variance method was used.

30-day mortality was analyzed through the risk ratio (RR) and its standard error (SE) using the inverse variance (IV) method. Finally, long term survival data were estimated from Kaplan-Meier curves using method described by Tierney et al. [10], where possible directly, and where this was not possible, by indirect methods, based on summary statistics.

Statistical significance was considered if *p* < 0.05 and 95% confidence interval did not include 1 in both cases. The analysis was completed using Review Manager version 5.3 for Windows (The Cochrane Collaboration, Software Update, Oxford, UK).

## 3. Results

### 3.1. Eligible Studies

Following the above specified search strategy, and after screening of the retrieved records, 42 articles were obtained in full text for further analysis. After evaluation of all these publications in full text format, 15 more articles were excluded for different reasons. The details about the studies excluded including the reasons for the exclusion can be found on the OSF (https://osf.io/3ybeq/?view_only=9e9bfcec4dd443efb919a258fb490edf).

After completing the literature search protocol, 27 studies [11,12,13,14,15,16,17,18,19,20,21,22,23,24,25,26,27,28,29,30,31,32,33,34,35,36,37] with the total of 4452 patients were found to meet the inclusion criteria and were included in the analysis following thorough assessment (Figure 1).

Out of these, 23 studies [11,12,13,14,15,16,17,18,19,22,23,24,25,26,27,28,29,30,31,33,34,35,36] *(n* = 2497) contained specific data about MR before and after surgery. Three publications [20,21,37] reported only mortality and survival data and one [32] was specific about pre and postoperative mitral anulus dimensions. The data referring to other than AS as an indication for AVR, or concomitant procedures undertaken, were excluded from the analysis. Also, from studies that had two arms we analyzed only the subgroup of patients meeting the inclusion criteria, provided there was enough information. The characteristics of included studies are presented in Table 1.

### 3.2. Risk of Bias

Each study was individually assessed for possible bias according to the before mentioned criteria. In the majority of studies there was a low risk of bias in terms of allocation concealment and blinding of participants and personnel as well as in the attrition and reporting bias. On the other hand, the risk of the detection bias–blinding of outcome assessment was considered high in the two thirds of publications. The random sequence generation bias was mostly marked as unclear due to the imprecise or uncertain data provided. The results of the risk of bias estimation are graphically presented (Figure 2).

### 3.3. Change in MR after Aortic Valve Replacement

Twenty-three studies mentioned before were included in the analysis of the MR. All of them used transthoracic echocardiography (TTE) for assessment, accompanied with transesophageal echocardiography (TOE) in three studies [18,31,32]. Only one study used TOE exclusively [25]. Substantial inter-study variation was noted in the method for reporting MR severity. Nevertheless, all of the approaches were in line with the 2010 European Association of Echocardiography recommendations for the assessment of valvular regurgitation. To consolidate the results across studies, all patients with precise data about MR (*n* = 1860) were stratified according to the severity of MR into four groups: nil (0), mild (1+), moderate (2+), severe (3+), both before and after the surgery. There were no patients with 4+ mitral regurgitation. The first comparison was conducted with severe MR as the outcome. It showed substantial decrease in MR severity postoperatively (RR, 1.65; 95% CI 1.36–2.00; *p* < 0.00001; I² = 40%) without significant inter-study heterogeneity. Subgroup analysis considering younger and older patients showed that the effect was more expressed in older patients (1.93; 95% CI 1.47–2.54; *p* < 0.00001) vs. younger (1.35; 95% CI 1.03–1.77; *p* = 0.03) (Figure 3).

The decrease in MR severity postoperatively was also evident with composite outcome of moderate/severe MR (RR, 2.11; 95% CI 1.41–3.15; *p* = 0.0003; I² = 98%), although this time with marked inter-study heterogeneity. Subgroup analysis considering younger and older patients showed that the effect was significant only in older patients (2.32; 95% CI 1.70–3.16; *p* < 0.00001) but not in younger patients (1.83; 95% CI 0.97–3.48; *p* = 0.06) (Figure 4).

The decrease in MR was statistically significant in both comparisons. Fifteen studies [11,12,14,19,22,23,25,26,28,29,30,31,33,35,36] (*n* = 1375) reported data about individual direction of MR change. An overall improvement in MR postoperatively was seen in 56.1%, 35.3% remained unchanged and in 8.6% the condition was aggravated. The average change in the degree of MR after AVR was reported or was possible to calculate in all 23 studies. After pooling and analyzing these data it was shown that, after the surgery, MR decreased on average by 0.46 (WMD; 95% CI 0.35–0.57; *p* < 0.00001; I² = 91%). However, the significant heterogeneity between studies was present.

The funnel plot analysis did not show significant asymmetry, so it was assumed that the publication bias was not present (Figure 5).

### 3.4. Reverse Myocardial Remodeling

The change in EF after AVR was reported in nine studies [15,17,18,22,24,27,30,34,36]. The analysis showed early increase in EF after surgery, but the result did not reach statistical significance (*p* = 0.14, WMD −1.02; 95% CI −2.36, 0.32; I² = 0%). There was no heterogeneity between studies. All these studies except one [27] also provided data about LVEDD, both before and after the intervention. Evidence of structural remodeling was demonstrated by significant reduction in the diastolic diameter of LV (*p* < 0.0001, WMD 0.21; 95% CI 0.12, 0.31; I² = 0%), which is shown in Figure 6. There was no heterogeneity between studies in terms of this parameter. Finally, three studies [17,18,32] measured the change in mitral annulus dimension. Pooling these results showed significant decrease in its diameter (*p* = 0.004, WMD 0.23; 95% CI 0.07, 0.38; I² = 0%), once again without significant heterogeneity (Figure 7). Further analysis of other echocardiographic parameters of reverse remodeling was impossible due to the lack of data reported.

### 3.5. Mortality and Long-Term Survival

The overall 30-day mortality was reported in seven studies [15,16,20,23,25,26,28]. Analysis of pooled data for these patients demonstrated a 5% relative risk of 30-day mortality after AVR. Significant heterogeneity between studies was not present. Five studies [16,20,23,30,37] compared 30-day mortality between groups of patients that had nil/mild versus moderate/severe grade of MR preoperatively. Combining these data revealed a significantly higher mortality rate in the moderate/severe group (*p* < 0.0001; I² = 0%, Figure 8.)

In the following five studies [16,19,20,21,26] the Kaplan-Meier long-term survival curves were presented. The data provided compared survival rates between groups based on preoperative MR severity. Analysis of pooled estimates showed significantly better survival of patients that had nil/mild versus moderate/severe MR preoperatively (*p* < 0.00001; I² = 0%), without inter-study heterogeneity. The data referred to the follow up period of 5 years (Figure 9).

## 4. Discussion

Aortic valve replacement continues to be one of the most often performed procedures in cardiac surgery, despite the advancement in new technologies, especially transcatheter valves. Also, it is one of the safest procedures, with observed mortality around 2.5%, less than expected, even with the ageing population [38]. Due to the fact that the population is getting older, besides higher incidence of the severe aortic stenosis that needs operative treatment, the number of conjoined medical conditions and comorbidities is increasing as well. Thus, the risk assessment and operative strategy planning becomes more complicated.

Looking for the literature dealing with this problem we found only five expert review articles addressing the topic descriptively without the proper meta-analysis and more importantly, without the clear conclusion. Almost all other publications were relatively small retrospective studies. So we decided to perform a meta-analysis of all published articles up to date.

There were no randomized controlled trials regarding this topic. After thorough literature search the majority of published research articles that met the criteria for inclusion in the meta-analysis were retrospective observational studies. Only in the three studies [18,30,34] the patients were prospectively included in the research, but still the lack of randomization, blinding and proper study protocol decreased the strength of the data. All of the publications based their research on the comparison of the parameters in the same patients before and after the intervention.

Taking all this into account we performed thorough examination of all included studies for the possible bias. The investigation returned somewhat expecting results. Namely, the Achilles heel in the majority of publications were random sequence generation on one side and the blinding of the outcome assessment on the other. As the studies were mostly retrospective in nature, they usually included all patients operated for aortic valve stenosis with some degree of concomitant mitral regurgitation that was left unaddressed. Depending on specifying the exact period of time and place of obtaining the patient records the risk of selection bias in terms of randomization was determined as high or unclear, but only for the prospective studies we estimated it as low although there was no real randomization, as previously said.

The risk of bias in the blinding of outcome assessment was even more prone to be labeled as high due to the obtaining of postoperative data in the retrospective search through medical records. These echocardiograms were probably performed by the different examiners through the long period of time and without the proper study protocol. In some of the publications [12,13,24,25,30,37] the authors stated that they retrieved the echocardiographic recordings for reevaluation, done by one or two researchers following the agreed protocol, so in these cases the risk of bias was determined as low. In his prospective study, Tassan-Mangina [18] stated that “postoperatively the same transthoracic and transesophageal parameter measurements were performed using the same methods” so there was a low risk of bias in the outcome assessment as well. Brasch and colleagues [14] reported that two cardiologists analyzed echo Doppler studies, but it was not clear whether it was a reevaluation following the study protocol or those were the original records so the risk of bias concerning the outcome assessment was labeled unclear. In all other studies there was no clear explanation about the obtaining of the outcome data or the data was collected from the past medical records so the risk of bias was designated as high.

Regarding the allocation concealment and blinding of participants and personnel the situation was much better. There was a low risk of bias in most researches since all patients underwent the same treatment—surgical aortic valve replacement.

Even more so was the case with the potential attrition bias or selective reporting. All but two studies were appraised very high for the outcome data provided. In the publication by Wan and colleagues [26] the problem was that the outcome data was not presented for all of the included patients. The outcome data was missing for 31 persons (16%). In that case there was potentially a very high risk of the selective reporting that could distort the results and conclusions. The problems with the follow up data were encountered also in the study by Takeda [28]. They even disclosed themselves that “the follow-up echocardiographic measurements were incomplete and opportunistic and there is potential for sampling bias in the assessment of MR at the late period.” So, apart from these two studies the reporting of the outcome data was considered generally fair and unbiased.

After pooling all the individual data of 1860 patients, the clear improvement of MR after AVR alone was demonstrated. Looking on the severe MR as the outcome, the decrease was significant (RR 1.65; 95% CI [1.36, 2.00]). Since there was no significant inter-study heterogeneity (I² = 40%) one could conclude that this effect is constant and predictable. Trying to evaluate this finding more precisely and to see whether this decrease in the MR can be expected also for the less pronounced cases we repeated the analysis but this time we put the threshold a little lower. The investigation was performed with the composite outcome of moderate + severe MR. In this scenario the improvement was even more pronounced (RR 2.11; 95% CI [1.41, 3.15]), although with marked heterogeneity between studies. The reasons for this heterogeneity could be differences in age, clinical status and sample size of the patient populations (Table 1). Nevertheless, the overall effect was the decrease in MR.

As this heterogeneity shown above could be the source of bias and could lead to the distortion of the results one should be careful with the conclusions. Seeking to overcome the potential source for this heterogeneity we tried to divide the studies into two subgroups according to the average age of the patient populations. In the “older” subgroup were 11 studies where the patients were older than 70 years on average [13,14,16,17,24,25,26,27,30,31,36] and in the “younger” subgroup the patients were younger than 70 [12,15,18,19,22,23,28,29,33,34,35]. This grouping showed that the decrease in MR was significant only in older patients, but not in the younger subgroup. As a matter of fact, this is a very important finding since the members of the elderly population are much more often the subject of the key question, to operate on the mitral valve or to leave it. Whether this finding comes from the situation that younger patients have more time to gradually develop the worsening of the initially subsided MR after the AVR or there are some different reasons for this remains to be answered. But, considering the usually higher perioperative risk in older people the evidence of the decrease of MR suggests that we should be more conservative in the operative planning [39]. Shorter life expectancy in this subgroup of patients only supports the decision only to replace the aortic valve and to leave the moderate MR behind as it would decrease to a certain degree.

What about the magnitude of this effect? After combining all data about the degree of MR change across all 23 studies it was shown that MR decreased for 0.46 on average. In seven studies with the least marked improvement [11,15,18,19,22,31,33] (*n* = 641) this decrease was negligible 0.17 [0.11, 0.24]. Seven studies with the highest levels of MR improvement [13,23,26,27,28,29,34] (*n* = 706) have shown MR decrease of 0.75 [0.66, 0.83].

Observing again the individual data across all patients we found that this improvement was actually seen in 56.1% while the majority of others remained unchanged (35.3%). It is obvious that there is still a substantial number of patients with more than mild MR.

Fifteen studies were searching for predictors that influence the change in MR. The following parameters predicted the lack of MR improvement: atrial fibrillation (AF) [25,27], increased LV mass [14,18], coronary artery disease (CAD) [19,23] and higher grade of MR [24,25,27,35]. There was no clear predictor for the decrease in MR. Kaczorowski and colleagues examined the relation of MR change to preoperative aortic valve gradient but found no relationship.

Concerning reverse remodeling, from echocardiographic parameters reported, besides EF, we found enough data to analyze only LVEDD and mitral annulus dimensions pre and postoperatively. As expected, after the AVR there was an increase in EF. However, this increase did not reach statistical significance, probably due to the short period of time for reverse remodeling to fully develop. The majority of studies did not report the exact time of the postoperative echocardiography exam. The effect was the same when calculated only for three studies [24,27,34] that reported entering mean EF under 50%. On the other hand, when pooling the data about LVEDD, a significant decrease post AVR was shown. The evidence of structural remodeling taking place was further demonstrated after meta-analysis of three studies that measured mitral annulus dimensions. There was a significant decrease in its diameter post AVR. The deeper analysis of the reverse remodeling was not possible due to insufficient parameters reported, but it is obvious that it is taking place and favorable effects on MR could be expected.

It is already said that surgical AVR is a relatively safe procedure with unadjusted perioperative mortality rate under 3%. For the patients with concomitant moderate MR our meta-analysis combining the data from seven studies showed that there was a 5% relative risk of 30-day mortality after AVR alone. There were also five studies that compared mortality between the groups based on the presence of MR preoperatively and pooling all the data confirmed significantly higher perioperative mortality in patients with MR (*p* < 0.0001, Figure 8). So, their risk was nearly double but still less than half of the mortality observed when double valve surgery was undertaken. In the follow up period this initial difference in mortality persisted and their survival curves diverged. Comparing Kaplan-Meier curves showed that patients with MR had significantly worse survival after 5 years of follow up. In regard to this suboptimal final outcome we think that the intervention on mitral valve at the time of AVR is strongly recommended.

## 5. Limitations

All papers published on this issue are observational, usually retrospective studies. There are no randomized trials. Apart from study design, inter-study variations in patients’ demographics could also influence the results. And finally, although echocardiography parameters were reported following European Association of Echocardiography guidelines, there was substantial difference between studies that sometimes forced us to exclude certain data from the analysis. Anyway, these are the best data we have until more rigorous prospective studies if not randomized trials emerge.

## 6. Conclusions

The moderate MR should be addressed at the time of AVR. The only exception, or reason to leave it behind could be for the elderly population and the minority of very high risk patients with a lot of comorbidities.

## Figures and Tables

**Figure 1 ijerph-17-07335-f001:**
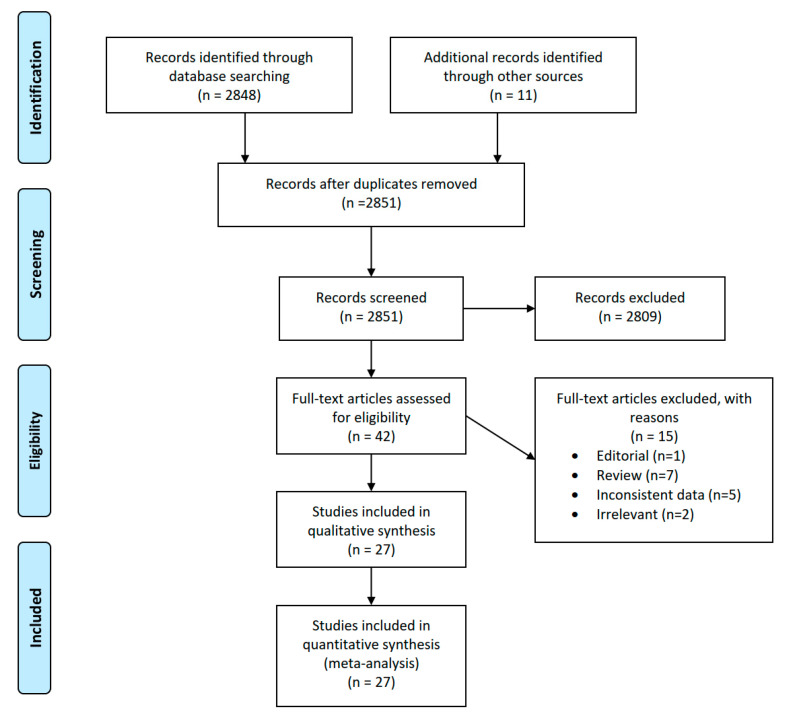
PRISMA flow chart of literature search.

**Figure 2 ijerph-17-07335-f002:**
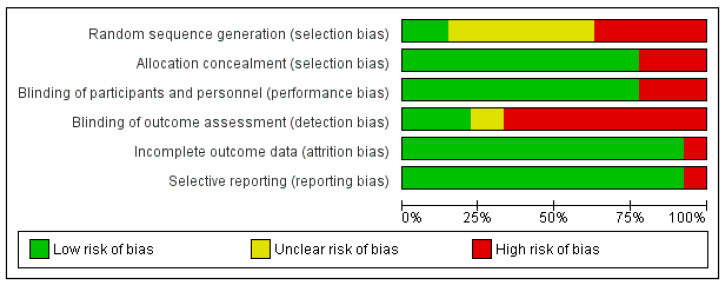
Risk of bias–estimation across studies.

**Figure 3 ijerph-17-07335-f003:**
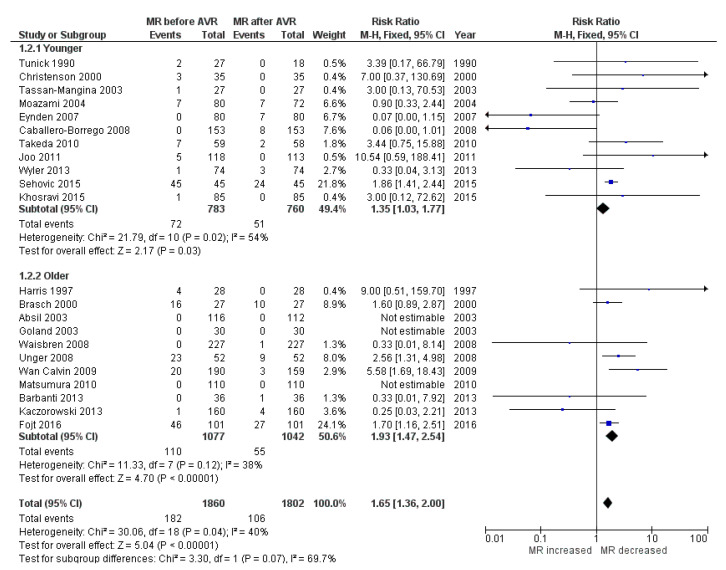
The forest plot of risk ratio of MR after AVR. The outcome: severe MR.

**Figure 4 ijerph-17-07335-f004:**
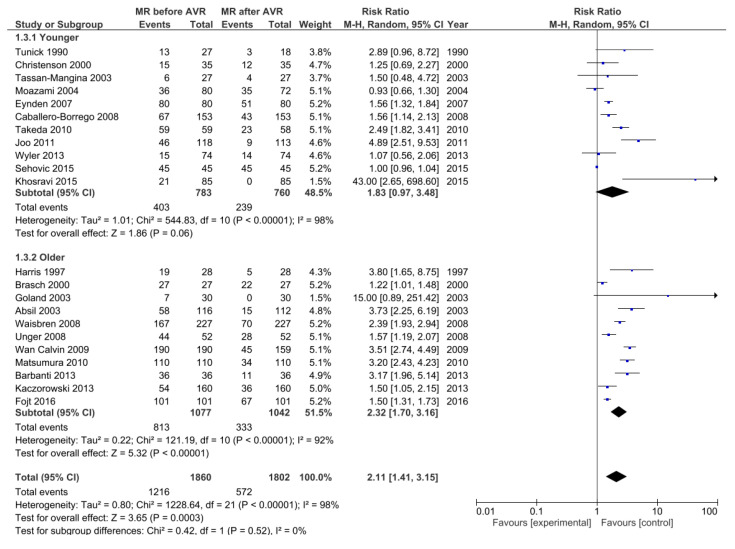
The forest plot of risk ratio of MR after AVR. The outcome: moderate/severe MR.

**Figure 5 ijerph-17-07335-f005:**
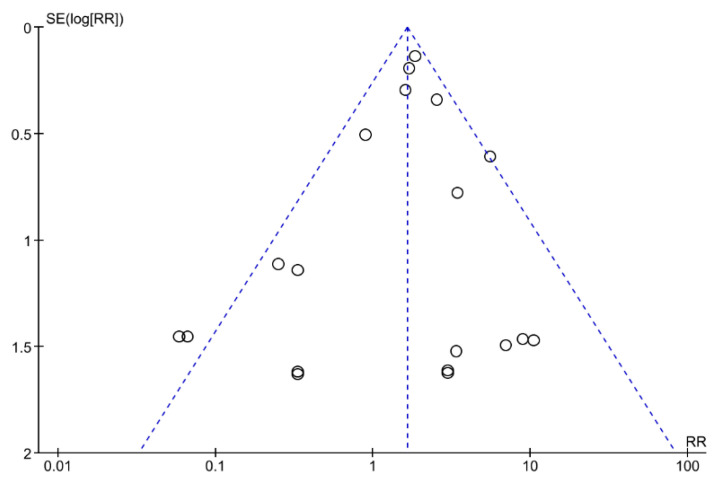
Funnel plot analysis for publication bias.

**Figure 6 ijerph-17-07335-f006:**
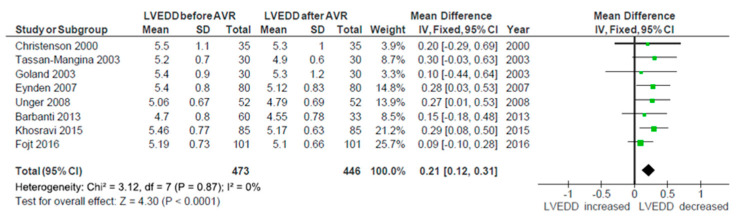
The change in LVEDD after AVR.

**Figure 7 ijerph-17-07335-f007:**
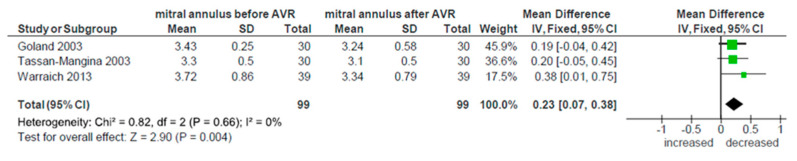
Mitral annulus diameter change after AVR.

**Figure 8 ijerph-17-07335-f008:**
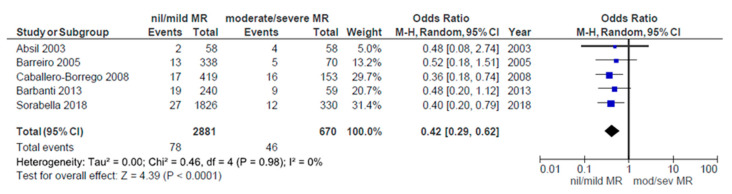
Comparative 30-day mortality: nil/mild vs. moderate/severe MR groups.

**Figure 9 ijerph-17-07335-f009:**
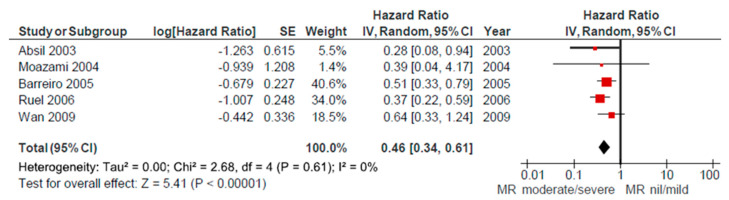
5-year survival comparison: nil/mild vs. moderate/severe MR groups.

**Table 1 ijerph-17-07335-t001:** Characteristics of studies included in the meta-analysis.

Study Reference	Year	Design	Total Pts n	Age	NYHA Mean	Mean Aortic Gradient	MR Measuring and Reporting	MR Change Average	Significance of MR Change
Adams et al. [11]	1990.	retrospective	24	66 ± 15		63.0	semiquantitative TTE	NS	NS
Tunick et al. [12]	1990.	retrospective	44	69 ± 12			semiquantitative TTE, pulsed and continuous wave doppler	MR grade −0.39	*p* < 0.05
Harris et al. [13]	1997.	retrospective	28	75 ± 8		42.0	semiquantitative TTE, pulsed, continuous and color doppler	MR jet area −3.0 cm²	*p* < 0.0001
Brasch et al. [14]	2000.	retrospective	27	77 ± 17		36.0	semiquantitative TTE, color dopler	MR grade −0.6	*p* = 0.005
Christenson et al. [15]	2000.	retrospective	36	64.0 ± 13.3	3.0		semiquantitative TTE, color dopler	NS	NS
Absil et al. [16]	2003.	retrospective	116	74.8 ± 7.1			semiquantitative TTE, pulsed and continuous wave doppler	MR grade −0.37	N/A
Goland et al. [17]	2003.	retrospective	30	72.0 ± 6.5			TTE, color dopler, indexed jet area	MR grade −0.5	*p* = 0.0012
Tassan-Mangina et al. [18]	2003.	prospective	30	68 ± 8		55.0	Color dopler jet area TTE, TOE	MR grade −0.15	*p* = 0.016
Moazami et al. [19]	2004.	retrospective	107	67.1	2.7		semiquantitative Color dopler, TTE	NS	NS
Barreiro et al. [20]	2005.	retrospective	408	78.1 ± 5.4			semiquantitative TTE, jet area range	N/A	N/A
Ruel et al. [21]	2006.	retrospective	848	69.6 ± 11.6			semiquantitative TTE, color dopler	N/A	N/A
Van den Eynden et al. [22]	2007.	retrospective	80	66 ± 11	2.8	50.3	Quantitative TTE	MR grade −0.27	*p* < 0.0001
Caballero-Borrego et al. [23]	2008.	retrospective	153	68.3 ± 9.2		54.5	TTE, color dopler, regurgitant jet area	MR grade −0.67	*p* < 0.0001
Unger et al. [24]	2008.	prospective	52	77		42.0	Quantitative TTE	Rvol −8.3 mL	*p* < 0.0001
Waisbren et al. [25]	2008.	retrospective	227	71 ± 11		51.0	Quantitative TOE	MR grade −0.43	*p* < 0.0001
Wan et al. [26]	2009.	retrospective	190	74 ± 11	3.0	54.0	semiquantitative TTE, color dopler	MR grade −0.80	*p* < 0.0001
Matsumura et al. [27]	2010.	retrospective	110	73 ± 10		43.0	Quantitative TTE	MR jet area −3.3 cm²	*p* < 0.001
Takeda et al. [28]	2010.	retrospective	59	67 ± 11			semiquantitative TTE, color dopler	MR grade −0.76	*p* = 0.003
Joo et al. [29]	2011.	retrospective	118	63 ± 13	2.7		semiquantitative TTE, color dopler	MR grade −0.98	significant
Barbanti et al. [30]	2013.	prospective	299	86.2 ± 5.9	3.5	42.7	semiquantitative TTE, color dopler	MR grade −0.47	significant
Kaczorowski et al. [31]	2013.	retrospective	462	72.9 ± 10.9	2.7	44.6	semiquantitative TTE, TOE, color dopler	MR grade −0.28	*p* < 0.05
Warraich et al. [32]	2013.	retrospective	39	74.0 ± 13.7			Quantitative TOE, TTE	N/A	N/A
Wyler et al. [33]	2013.	retrospective	74	68.7 ± 11.7	2.6	54.9	semiquantitative TTE, color dopler	NS	NS
Khosravi et al. [34]	2015.	prospective	85	56 ± 6.1		35.6	semiquantitative TTE, color dopler	MR grade −0.64	*p* < 0.01
Sehovic et al. [35]	2015.	retrospective	45	56.25 ± 7.24	2.2		Quantitative TTE	MR grade −0.47	significant
Fojt et al. [36]	2016.	retrospective	101	76.1 ± 8.2	2.6	42.4	semiquantitative TTE, color dopler	MR grade −0.55	*p* < 0.001
Sorabella et al. [37]	2018.	retrospective	660	78.1 ± 10.1			semiquantitative TTE, color dopler	N/A	N/A

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
