# Peer review of "The Prospects of Secondary Moderate Mitral Regurgitation after Aortic Valve Replacement —Meta-Analysis"

_ijerph, 2020, doi:10.3390/ijerph17197335_

Round 1

Reviewer 1 Report

In this article, authors demonstrated, using meta-analysis, the change of preoperative MR and its impact on outcomes after aortic valve surgery to suggest treatment strategy of aortic stenosis with concomitant MR. This topic is still of clinical interest and no robust data have supported whether the non-organic MR should be treated or left at the aortic valve surgery. Although the paper is well-written, please consider the following:

Comments

  • Introduction, Line 70; authors state that the objective of the study is to create treatment guidelines for AS with concomitant MR of a moderate degree. However, authors showed the results of the patient group with severe or severe/moderate MR. For this purpose, authors should only focus on patients with moderate MR.
  • Introduction, Line 75; authors provided the question “does the expected reverse remodeling of the left ventricle have a beneficial effect on mitral regurgitation?”, but did not show any data to answer it at all.
  • Introduction, Line 79; primary endpoint was focused only on secondary MR patients. Thus, authors should describe how authors excluded studies involving organic MR in the methods section or in Figure 1.
  • Results, Line 22; subgroup analysis was performed divided by age. However, authors just describe the groups as younger or older without providing any cut-off value. The clear definition of younger and older should be included in the methods or results section.
  • Discussion; in the first several paragraphs of discussion section, authors put similar discussion to introduction section. Thus, it is recommended shorten introduction section by deleting them. Introduction is a bit redundant and should be simplified to clearly describe the background and aims of this study.
  • Discussion, Line 192; authors state that for the patients with concomitant moderate MR, meta-analysis showed a 5% relative risk of 30-day mortality after AVR alone. But, this number seems to include severe MR patients when looking at the data in the result section.

Author Response

  1. As we noted in Results, line 18/19, there were no patients with severe 4+ MR because that finding would indicate the need for mitral surgery as well. For the sake of easier understanding we designated the stratified groups with MR 1+ as “mild”, MR 2+ as “moderate” and MR 3+ as “severe”. We did it to overcome the potential confusion if the MR 3+ is designated as “moderate/severe” what is usually the case.
  2. As mentioned in the Results under separate section, from the line 46, concerning remodeling, we were able only to analyze the change in EF and mitral annulus diameter as reverse remodeling marks. There were not enough data provided in the publications to check its specific impact on mitral regurgitation.
  3. We failed to mention in inclusion criteria that only the publications concerning secondary MR were included, but now it is corrected (Methods, line 95). If there were different groups of patients in one particular study, only the patients with secondary MR were included in the meta-analysis. (Methods, line 98).
  4. The cut-off value was 70 years of age. We accidentally missed out to explain it, but now it is corrected. (Methods, line 123)
  5. Thank you, we revised and corrected the introduction section.
  6. As mentioned before “severe MR” refers to MR 3+, commonly labeled as “moderate/severe”. There were no patients with 4+ MR.

Reviewer 2 Report

Authors performed systematic review and meta-analysis to reveal whether the moderate mitral regurgitation will improve after aortic valve replacement alone. there was significant improvement in mitral regurgitation postoperatively. The effect is more pronounced in the elderly population. Perioperative mortality was higher and long-term survival significantly worse in patients that had moderate/severe mitral regurgitation preoperatively.

Authors concluded that there is only slight improvement in concomitant mitral regurgitation after aortic valve replacement alone. Moderate mitral regurgitation should be addressed at the time of aortic valve replacement. The approach should be more conservative in the elderly and high risk patients.

This review was well designed and meta-analysis presented clear results, which reached their conclusion. I think this review article should contribute certain information for Journal readers. I have several comments as the followings;

  1. Some more information is required to reach the final conclusion that the approach should be more conservative in the elderly and high risk patients. Authors presented perioperative mortality according to severity of mitral regurgitation. I think authors should present perioperative mortality according to age and mitral regurgitation. There was little information about recommendation of conservative approach for elderly patients.
  2. Authors mentioned as the followings; Double valve surgery carries a significantly higher operative risk comparing to aortic valve replacement alone. Perioperative mortality of double valve surgery reaches 11-12 %, while that of AVR alone was 3.2%. Perioperative mortality of mitral valve replacement was about 5%, while that of mitral valve repair was about 1.2%. Authors should present perioperative mortality of AVR concomitant with mitral valve repair because mitral valve repair is major procedure for patients who underwent AVR.
  3. Authors analyzed 30-day mortality. I think perioperative mortality is better to use 30-day mortality.

Author Response

  1. As this article is a meta-analysis, we could only use the data provided in the publications dealing on the topic. I agree with you that there is not enough information to make clear recommendations but after pooling all the individual results from the studies there was statistically significant decrease in MR in the older patients. Considering their increased operative risk this information at least supports the decision not to operate on the mitral valve in the elderly.  
  2. I agree with you that the mitral valve repair concomitant with the AVR is a major procedure. That was the very point that led us to start this investigation. The reason we didn`t mention the exact mortality of that kind of procedure is because there is a wide difference in the data reported by the different institutions. But we believe that the perioperative mortality in that case approaches the one of double valve surgery. Since this is a meta-analysis we could only work with the data reported in the included publications.
  3. The articles included in the meta-analysis used different outcome measures. But the majority of them reported the 30-day mortality. As for the follow up there was even less information, but five studies reported the 5-year survival so we could include their data in the overall analysis. That is the reason why we reported only those mortality outcome measures.

Reviewer 3 Report

Dear Authors,

I have read submitted paper “The Destiny of Moderate Mitral Regurgitation After Stenotic Aortic Valve Replacement and Importance in The Elderly Population – Meta-Analysis” with great attention.  This is an interesting attempt to establish the optimal treatment strategy for patients referred for AVR with concomitant mitral regurgitation. The study design is sound, and the meticulous meta-analysis is well described, however the submitted manuscript would still benefit with some corrections:

  1. Most parts of the manuscript, except for Methods section (abstract, introduction, discussion and conclusions) need extensive linguistic edition, preferably by a native speaker.

  1. The title is not clear and should be modified.

  1. In the Introduction the Authors write: „The objective of this study is to try to create treatment guidelines”. The treatment guidelines are usually created by a dedicated Working Group, while the results of this meta-analysis can obviously be helpful for such purpose, however cannot be considered as guidelines itself. 

  1. Most of the papers included in the meta-analysis are retrospective analyses, however four of them were designed as prospective observational studies. Including both types in the submitted meta-analysis, although practical, should be considered as another limitation of the study and mentioned in the study limitation section accordingly.

  1. The papers included in the meta-analysis presented in Table 1 should be numbered and the reference position should be given for the sake of clarity.  

  1. The Discussion, although interesting, is somewhat chaotic and should be organized.

  1. The Conclusions section is too long. It is not a place for speculations and rhetorical questions. Only the conclusions supported by the presented data should be put here in a straightforward manner.

This comments do not affect the great value of your work, which I do appreciate.

Author Response

  1. Thank you for your suggestion. We appointed a professional English language lecturer for the linguistic editing.
  2. We modified the title to make it more precise and clear.
  3. Yes, actually, the objective of this study is to help in creating treatment guidelines. We corrected it.
  4. These limitations are specifically mentioned at the end of discussion, line 202
  5. We added the reference numbers in Table 1
  6. Thank you. We tried to modify the Discussion section to make it more clearly organized.
  7. We shortened the conclusions section and tried to make it clear.